# Feasibility of Determination of Foodborne Microbe Contamination of Fresh-Cut Shredded Cabbage Using SW-NIR

**Benjamaporn Matulaprungsan** [1,2], **Chalermchai Wongs-Aree** [1,3], **Pathompong Penchaiya** [3], **Phonkrit Maniwara** [4], **Sirichai Kanlayanarat** [1], **Shintaroh Ohashi** [2,5] **and Kazuhiro Nakano** [2,5,*]

[1] Postharvest Technology Program, School of Bioresources and Technology, King Mongkut's University of Technology Thonburi, Bangkok 10150, Thailand; Bee2bubble@gmail.com (B.M.); chalermchai.won@kmutt.ac.th (C.W.-A.); sirichai1076@hotmail.com (S.K.)

[2] Graduate School of Science and Technology, Niigata University, 8050, Ikarashi 2-no-cho, Nishi-ku, Niigata 950-2181, Japan; sohashi@agr.niigata-u.ac.jp

[3] Postharvest Technology Innovation Center, Office of the Higher Education Commission, Bangkok 10400, Thailand; doi32306@hotmail.com

[4] Postharvest Technology Research Center, Faculty of Agriculture, Chiang Mai University, Chiang Mai 50200, Thailand; maniwara016@gmail.com

[5] Environmental Science and Technology, Institute of Science and Technology, Niigata University, 8050, Ikarashi 2-no-cho, Nishi-ku, Niigata 950-2181, Japan

[*] Correspondence: knakano@agr.niigata-u.ac.jp

**Abstract:** Shredded cabbage is widely used in much ready-to-eat food. Therefore, rapid methods for detecting and monitoring the contamination of foodborne microbes is essential. Short wavelength near infrared (SW-NIR) spectroscopy was applied on two types of solutions, a drained solution from the outer surface of the shredded cabbage (SC) and a ground solution of shredded cabbage (GC) which were inoculated with a mixture of two bacterial suspensions, *Escherichia coli* and *Salmonella typhimurium*. NIR spectra of around 700 to 1100 nm were collected from the samples after 0, 4, and 8 h at 37 °C incubation, along with the growth of total bacteria, *E. coli* and *S. typhimurium*. The raw spectra were obtained from both sample types, clearly separated with the increase of incubation time. The first derivative, a Savitzky–Golay pretreatment, was applied on the GC spectra, while the second derivative was applied on the SC spectra before developing the calibration equation, using partial least squares regression (PLS). The obtained correlation (*r*) of the SC spectra was higher than the GC spectra, while the standard error of cross-validation (SECV) was lower. The ratio of prediction of deviation (RPD) of the SC spectra was higher than the GC spectra, especially in total bacteria, quite normal for the *E. coli* but relatively low for the *S. typhimurium*. The prediction results of microbial spoilage were more reliable on the SC than on the GC spectra. Total bacterial detection was best for quantitative measurement, as *E. coli* contamination could only be distinguished between high and low values. Conversely, *S. typhimurium* predictions were not optimal for either sample type. The SW-NIR shows the feasibility for detecting the existence of microbes in the solution obtained from SC, but for a more specific application for discrimination or quantitation is needed, proving further research in still required.

**Keywords:** shredded cabbage; SW-NIR; total bacteria; *E. coli*; *S. typhimurium*

## 1. Introduction

Public concern about the safety of fresh produce consumption has been continuously growing. The power of social media also helps to spread the news about foodborne disease infections faster than

ever. The awareness of producers is high when it comes to protect the contamination problem of their products. However, many foodborne disease outbreaks in fresh and fresh-cut produce are still being reported [1]. Fung et al. [2] found that *Staphylococcus*, *Salmonella*, *Clostridium*, *Campylobacter*, *Listeria*, *Vibrio*, *Bacillus*, and *Escherichia coli* accounted for 90% of global food poisoning illnesses. *E. coli* and *Salmonella* are two pathogenic bacteria mainly concerned with fresh-cut produce [3–6]. This report agrees with the food poisoning statistics from the Japanese Ministry of Health, Labour, and Welfare about the most common causes of foodborne disease in Japan in 2011 [7]. For fresh and fresh-cut producers, assuring the safety of the produce to the consumer is inescapable.

Shiina and Hasegawa [8] reported the fast multiplication of the fresh-cut produce companies in Japan is due to the high demand of the fresh-cut vegetables, which highly relates to the increase of consumer convenience needs. Among the many kinds of fresh-cut produce, shredded cabbage holds the highest proportion of the market share. The widespread use of shredded cabbage in Japanese cuisine can be observed in restaurants, dining rooms, fast food shops, retail markets, and home-cooking. Hence, producers must ensure the consumers that their products are safe from foodborne disease microorganisms. The conventional microbiological methods, such as total plate count and coliform count, are still conducted in companies as the most reliable method, even though a duration for 48 h or more is needed to obtain the results [9]. Recently, it was found that the time-consuming process of conventional microbiological methods could be replaced by using Petrifilm™ (3M Company, St. Paul, MN, USA), proved to be more convenient as compared to the traditional method [10], but it is still expensive for routine identification. Therefore, many researches attempted to develop reliable and rapid nondestructive methods to determine the microorganism contamination in fresh-cut produce.

Application of near infrared (NIR) spectroscopy has been widely used as an analytical tool for measuring quality attributes of horticultural produce [11], and for controlling the quality of agro-food products [12]. NIR spectroscopy is fast, reliable, and non-destructive, which could help reduce the analytical time of the traditional methods. For the microbial determination, NIR spectroscopy is an alternative technique for classification and prediction of microorganism in isolated systems [13–15]. Furthermore, some reports previously utilized short wavelength-near infrared (SW-NIR) techniques for determining the quantity of total bacteria in real food samples, such as chicken [16], flounder fillet [17], raw milk [18] and shredded cabbage [9]. These results suggest that NIR is a promising potential technique for monitoring and evaluating the development of microorganisms in real food products. Although the quantitative detection of total bacteria was mostly reported, the qualitative analyses of specific types of bacteria were scarce. Cámara-Martos et al. [19] attempted to use FT-NIR spectroscopy for evaluating the difference between two bacterial species, *E. coli* and *Pseudomonas aeruginosa*, inoculated in ultra-high-temperature (UHT) processed whole milk. The results revealed that NIR could be applied to detect and quantify the bacteria in milk, but further study is needed for increasing the discrimination capacity of NIR. Therefore, the objective of this study is to evaluate the feasibility of using SW-NIR for detecting total bacteria, *E. coli* and *S. typhimurium*, in two different types of solutions obtained from fresh cut shredded cabbage samples.

## 2. Materials and Methods

### 2.1. Preparation of Shredded Cabbage

Cabbage heads were purchased from a local market and then transported to the NIR laboratory at Kasetsart University, Kamphaeng Saen campus, Nakhon Pathom province, Thailand. Three layers of outer leaves of each cabbage head were removed before washing in 50 mg·L$^{-1}$ sodium hypochlorite solution for 5 min. Then, the cabbage heads were cut into quarters by an alcohol-sterilized sharp knife, and then sliced into approximately 3-mm width fresh-cut shredded cabbage. The shredded cabbage was washed again in 50 mg·L$^{-1}$ sodium hypochlorite solution and drained with a colander prior to packing into the sterilized polyethylene bags.

## 2.2. Preparation of Bacterial Suspension

Strains of *E. coli* TISTR527 and *S. typhimurium* TISTR1469 were obtained from the Culture Collection of Thailand Institute of Scientific and Technological Research (TISTR). The strains were sub-cultured on nutrient broth (NB; Media laboratory, Mumbai, India) at 37 °C for 24 h. Growth of bacterial suspension was harvested when the absorbance of a sample measured at a wavelength of 600 nm (OD600) was at 0.4 in sterile saline was obtained. Each culture was centrifuged at 6000 rpm for 20 min, and the cell pallet was washed twice with a sterilized 0.85% saline solution and then resuspended in 1 mL of a sterilized 0.85% saline solution. Then, the cell suspension of *E. coli* and *S. typhimurium* were gently mixed in a 1:1 ratio and diluted in distilled water to achieve a concentration of about 3–7 log CFU·mL$^{-1}$. Finally, a cell suspension of both strains was inoculated into the solution from the shredded cabbage sample.

## 2.3. Bacterial Inoculum Procedures

The sample preparation procedure is illustrated in Figure 1. Shredded cabbage was separated into two groups, non-ground and ground samples. For the non-ground sample, 10 g of shredded cabbage was weighted and packed into a sterilized polypropylene (PP) plastic bag. Then, 2 mL of bacterial suspension was inoculated into each bag and gently mixed. All bags were incubated at 30 °C for 30 min, and 10 mL of sterilized 0.85% saline solution was added into each bag to wash the surface of the sample, and then drained. The drained solution of the shredded cabbage from each bag was collected and incubated at 37 °C for further analysis. For the ground sample, shredded cabbage was ground using a commercial blender for around 30 s to 1 min, and 10 g of the ground sample was weighted. The ground sample was transferred into a sterilized PP bag and 2 mL of bacterial suspension was added. Following this, all bags were incubated at 30 °C for 30 min, prior to the adding of 10 mL of sterilized saline solution and mixed. The mixed sample of ground cabbage and bacterial suspension (GC) was incubated at 37 °C for further analysis. Sampling was done in SC and GC at 0, 4, and 8 h, for the NIR spectra acquisition and microbial growth analysis. According to the final mixture, the SC represented the microbiological composition on the surface of the shredded cabbage, while GC represented both the surface and inside.

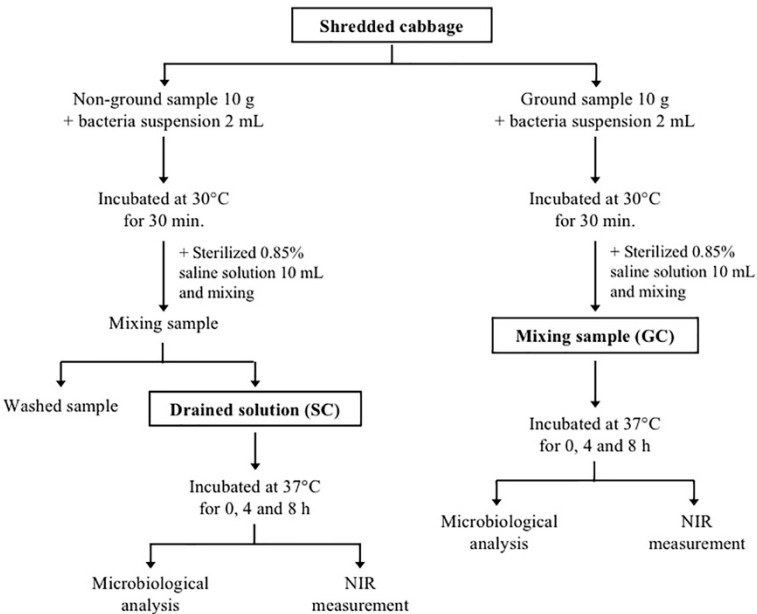

**Figure 1.** Sample preparation procedure.

## 2.4. SW-NIR Spectrum Acquisition

SW-NIR spectra was measured using a portable NIR spectrometer (FQA-NIR GUN, Feinted, Japan), producing a short wavelength region of 700–1100 nm, with a spectral resolution of 2 nm in reflection mode. To obtain the spectra from solution samples, the measuring cell was equipped with an aluminum block test tube holder. A Pyrex®-glass test tube [20 (∅) × 150 (h) mm] was used as the loading sample cell for the SC solution. As for the GC, a 30-mL Pyrex-glass beaker was used for the GC mixture solution and covered with a black cloth during measurement. Two spectra were recorded from two positions in each sample. The measurement was conducted at room temperature (25 °C). There were 72 spectra in total for the SC, and 69 for the GC samples.

## 2.5. Microbial Analysis

The microbial growth analysis was carried out in the SC solution and GC mixture solution after the NIR measurement. One percentage sterile peptone (100 mL) was added either to 10 mL of SC or to 10 g of GC samples and then mixed well by hand shaking for SC or stomacher (IUL Instruments Masticator, Barcelona, Spain) for GC. Each sample was comprised of 2 replications, each replication was managed for 3 dilutions, and each dilution was divided into 2 duplications. An aliquot (100 µL) of each duplication dilution was spread onto a petri dish of plate count agar and incubated at 37 °C for 24 h for the determination of aerobic mesophilic bacteria. While the culture of the eosin methylene blue agar was incubated at 37 °C for 24–36 h for *E. coli* counts, that of the xylose lysine deoxycholate agar was incubated at 37 °C for 24–48 h for the examination of *S. typhimurium*. These media are the standard method for enumerating our interested bacteria [20]. The number of bacteria were expressed as log CFU·mL$^{-1}$ for the SC solution, and as log CFU·g$^{-1}$ for the GC mixture solution.

## 2.6. Data Analysis

The CA maker software (Shizuoka Shibuya Seiki, Hamamatsu, Japan) was used to acquire the SW-NIR spectra from the portable NIR spectrometer. The spectra were further analyzed with the Unscrambler®software (CAMO, Oslo, Norway). The original spectra from the different sample preparation methods (SC and GC) and incubation times (0, 4, 8 h) were compared. The pretreatment of the first and second derivatives, by means of the Savitzky–Golay method, were applied to the original spectra. The principle component analysis (PCA) was initially conducted to observe the discrimination of the data set. Then, the pretreatment spectra were analyzed using the partial least square regression (PLS) in full-cross validation method. The optimum pretreatment method for each data set was chosen from the lowest standard error of cross-validation (SECV) and the highest correlation coefficients (*r*). Following this, the performance of a prediction model was considered from a set of validation samples, which were based on the statistical parameters as bias [21]. The significance of the bias was verified with a *t*-test using the following formula:

$$T_b = \pm \frac{t_{(1-\frac{\alpha}{2})SECV}}{\sqrt{n}}$$

where $\alpha$ is the probability of marking a type I error, *t* is the appropriate student *t*-value for a two-tailed test with degrees of freedom associated with the SECV and the selected probability of a type I error, *n* is the number of independent samples, and SECV is the standard error of cross-validation. If the bias value is lower than $T_b$, the bias is not significantly different from zero. The results of the bacteria analysis for both NIR measurement and total plate count, from the 72 spectra (SC) and 69 spectra (GC), were used for the construction model. Some samples were removed when detected as outliers. To evaluate the efficiency of the calibration model, the residual predictive deviation (RPD), ratio of

performance to inter-quartile range (RPIQ), and range error ratio (RER) were applied, by following these equations below:

$$RPD = SD/SECV \quad RPIQ = IQ/SECV \quad RER = Reference \; Range/SECV$$

where, SD is standard deviation, SECV is the standard error of cross-validation, IQ is an inter-quartile range: the difference between the values helps find 75% (Q3) and 25% of the samples (IQ = Q3–Q1), and the reference range is the difference between the maximum and minimum.

## 3. Results and Discussion

### 3.1. Microbiological Analysis

The descriptive statistics of the concentrations of total bacteria, *E. coli*, and *S. typhimurium*, from the different sample preparations, are shown in Table 1. Samples were collected from the SC and GC solutions mixed with a bacteria suspension of *E. coli* and *S. typhimurium* at 0 (before inoculation), 4, and 8 h (after inoculation) during the incubation period at 37 °C. At 0 h (before inoculation), the total bacteria detected in the SC solution was at 2.88 log CFU·mL$^{-1}$ and in the GC mixture solution at 3.15 log CFU·g$^{-1}$, while *E. coli* or *S. typhimurium* were not detected. After 8 h, the samples obviously changed from clear to an unclear solution indicating the microbial growth. The population of total bacteria increased to approximately 6–7 log CFU·mL$^{-1}$ (in SC) and log CFU·g$^{-1}$ (in GC). This level was above the microbiological limits for ready-to-eat food, 6 log CFU for total bacteria, 2 log CFU for *E. coli*, and nothing was detected in the 25 g of sample for *S. typhimurium* [22].

**Table 1.** Descriptive statistics of microbial growth, total bacteria, *E. coli* and *S. typhimurium*, in two types of samples; drained solution of shredded cabbage (SC) and mixed solution of ground cabbage and microbial suspension (GC).

| Sample | Bacteria | Microbial growth for SC in log CFU·mL$^{-1}$/ for GC in log CFU·g$^{-1}$) | | | | | |
| --- | --- | --- | --- | --- | --- | --- | --- |
| | | Min | Max | Mean | SD | IQ1 | IQ3 |
| | Total bacteria | 2.88 | 7.11 | 5.29 | 1.09 | 4.45 | 5.98 |
| SC | *E. coli* | 0.00 | 6.84 | 4.66 | 1.62 | 3.41 | 6.27 |
| | *S. typhimurium* | 0.00 | 6.18 | 3.53 | 1.26 | 3.00 | 4.29 |
| | Total bacteria | 3.15 | 7.06 | 5.35 | 1.03 | 4.58 | 6.25 |
| GC | *E. coli* | 0.00 | 6.59 | 4.60 | 1.88 | 2.85 | 6.00 |
| | *S. yphimurium* | 0.00 | 6.50 | 4.06 | 1.42 | 2.88 | 5.24 |

Remarks: SD: Standard deviation; IQ1: the first quartile, the value below which we can find 25% of the samples; IQ3: the third quartile, the value below which we can find 75% of the samples.

### 3.2. SW-NIR Spectra Analysis

The average SW-NIR spectra of the SC and GC, at different incubation times (0, 4 and 8 h), are shown in Figure 2a,b. For the SC spectra, each spectrum shifted upward with the increase in incubation time. For the GC spectra, each spectrum shifted downward with the increase in incubation time. This spectrum shift was also previously reported by Suthiluk et al. [9]. They suggested the growth of bacteria in the solutions, and the particles leaking from the cabbage cells were the cause of this spectrum shift. The solutions became less transparent, which alters the light-scattering properties [23]. The SC spectra separated nicely for each incubation time, while the GC spectra at 0 h was well separated from the spectrum at 4 and 8 h, which nearly overlapped. These results suggest that the SC solution had more potential for differentiating the quantity of contaminated microorganisms. However, the scattering effect was obviously found in the baseline shift of the spectra, from both the SC and GC solutions. Some pretreatment techniques, such as normalization or derivative Savitzky–Golay, have been recommended for removing this effect [24]. The second derivative Savitzky–Golay, was applied

to the SC spectra and the first derivative Savitzky–Golay on the GC spectra. Both methods successfully removed the baseline shift from the original spectra, as shown in Figure 2c,d. The strong absorption band at 962 nm due to the high amount of water in the solution was observed in both spectra. These spectra were then used for calibration.

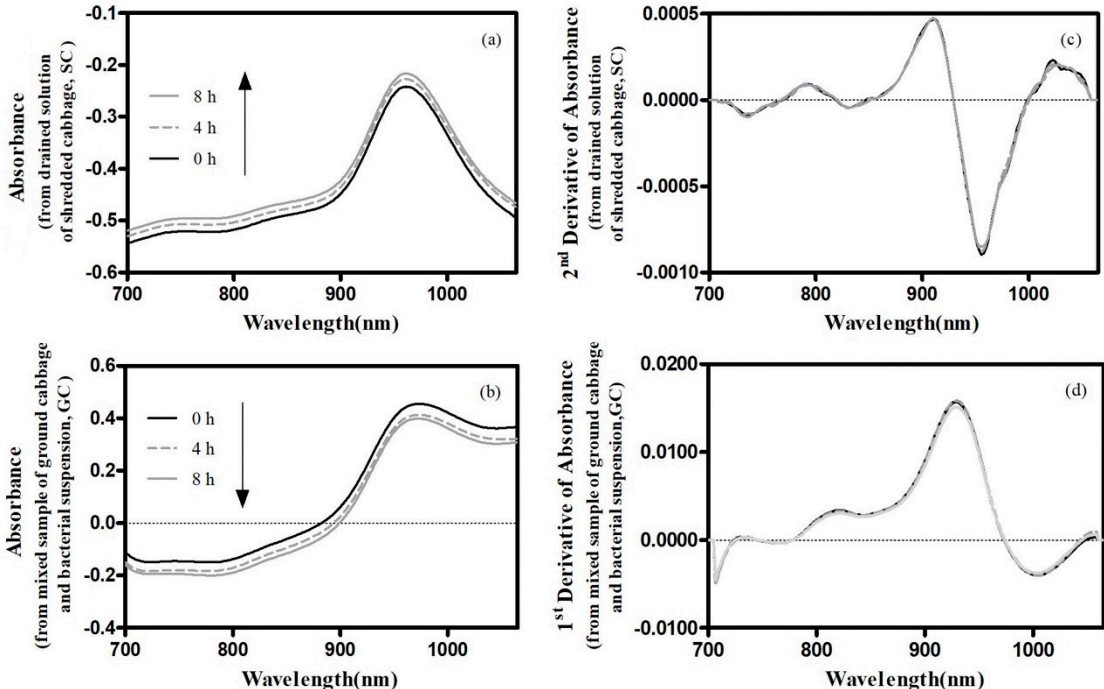

**Figure 2.** Original SW-NIR spectra of drained solution of shredded cabbage (SC) (**a**) and mixed solution of ground cabbage and bacterial suspension (GC); (**b**) at different incubation times. The second derivative Savitzky–Golay pretreated spectra of drained solution from shredded cabbage (SC); (**c**) and the first derivative Savitzky–Golay pretreated spectra from mixed solution of ground cabbage and bacterial suspension (GC) (**d**).

### 3.3. Feasibility of SW-NIR Used for Bacterial Detection

The principal component analysis (PCA) was applied to the SC and GC spectra treated with the Savitzky–Golay derivative, as mentioned earlier. PCA was applied to observe the clustering in the sample spectra. The score plot of the PCA of the SC and GC spectra, at different incubation times, are shown in Figure 3. The first two PCs (PC1 and PC2) accounted for over 90% of the total variance. The score plot of the SC samples between 0 h and 4 h, and 0 h and 8 h were completely segregated. Conversely, the score plot of 4 h and 8 h overlapped, indicating that the discrimination of the quantitative results may not be reliable. Meanwhile, the score plot of the GC samples showed inconsistency in the results. The clustering between 0 h and 4 h separated clearly but not for 0 h and 8 h, while the score plot of 4 and 8 h was well separated. This could be related to the transparency of the samples. SC samples were more transparent than GC samples, which contained both ground shredded cabbage particles and bacteria suspension. A better NIR application is highly related to the concentration of the chemical compound in the sample. The high counts of bacterial resulted in the good separation between samples, as previously reported by Rodriguez-Saona et al. [25] and Alexandrakis et al. [26].

The PLS models for total bacterial were developed on the pretreatment SC and GC spectra, and the results are shown in Table 2. The score plot of the predicted and measured value of total bacteria, are shown in Figure 4. The value of r of total bacteria detection from SC was best at 0.91, which showed a reliable prediction of 0.91–0.95. Followed by *E. coli* of the SC samples, the value was at 0.86, considered

as grading with approximation (0.81–0.90). The values of *E. coli* (0.79) and total bacteria (0.74) of GC samples and *S. typhimurium* (0.71) of SC samples could be rough-screened (0.70–0.80). The lowest *r* value was for *S. typhimurium* (0.47) from the GC sample, which indicates it is unusable (below 0.70). The r values of this study were based on the guidelines for interpretation of *r* [27]. The statistical result of the significance of bias for model performance ($T_b$) was higher than the biases for all parameters. $T_b$ values suggest that bias of developed models was considered as being no different from zero.

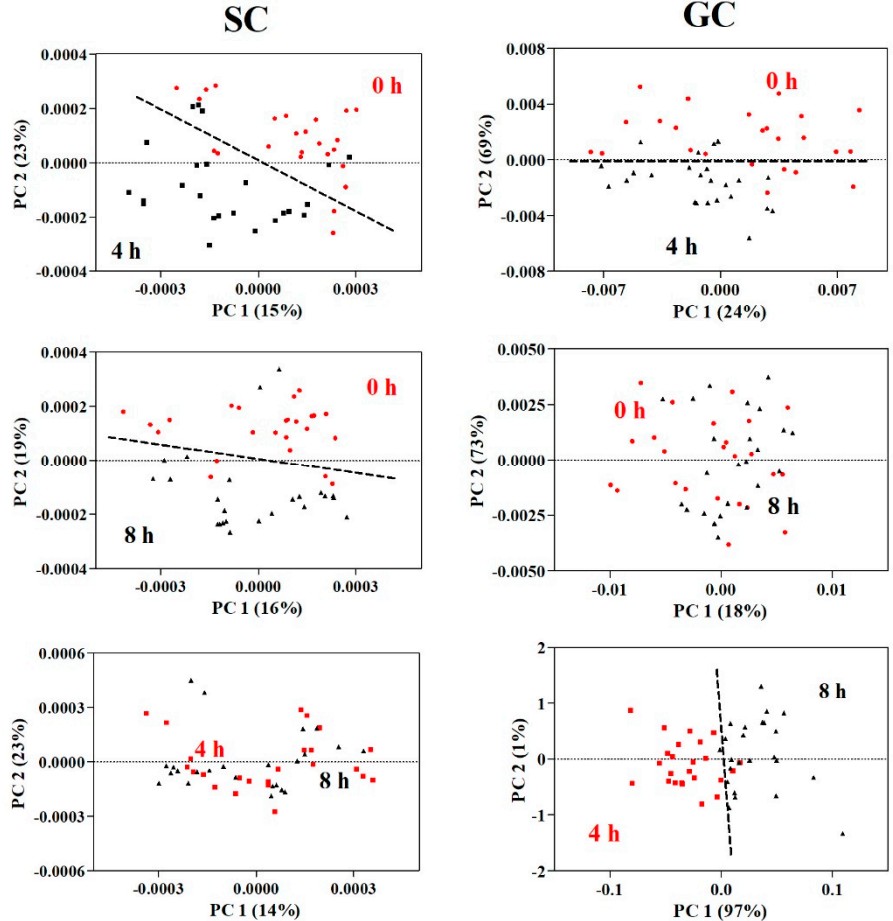

**Figure 3.** The principle component analysis (PCA) score plot (PC1 and PC2) of the pretreatment SW-NIR spectra of drained solution of shredded cabbage (SC, **left**) and mixed sample of ground cabbage and bacterial suspension (GC, **right**), in comparison between different incubation times, 0 h with 4 h, 0 h with 8 h, and 4 h with 8 h.

**Table 2.** PLS model result for predicting the microbial contamination in the drained solution of shredded cabbage (SC) and in the mixed sample of ground cabbage and bacterial suspension (GC).

| Sample | Bacteria | N | r | SECV | Bias | $T_b$ | RPD | RPIQ | RER |
|---|---|---|---|---|---|---|---|---|---|
| | Total bacteria | 72 | 0.91 | 0.45 | −0.02 | 0.10 | 2.44 | 3.40 | 9.55 |
| SC | *E. coli* | 72 | 0.86 | 0.83 | −0.12 | 0.20 | 1.95 | 3.45 | 8.18 |
| | *S. typhimurium* | 72 | 0.71 | 0.93 | −0.01 | 0.22 | 1.36 | 1.39 | 6.72 |
| | Total bacteria | 69 | 0.74 | 0.72 | −0.01 | 0.17 | 1.44 | 2.32 | 5.48 |
| GC | *E. coli* | 69 | 0.79 | 1.17 | −0.05 | 0.28 | 1.61 | 2.70 | 5.67 |
| | *S. typhimurium* | 70 | 0.47 | 1.28 | −0.02 | 0.31 | 1.11 | 1.84 | 5.10 |

Remarks: N: number of samples; *r*: correlation coefficient; SECV: standard error of cross validation; $T_b$: the statistic test of the significance of bias for model performance; RPD: ratio of performance to deviation; RPIQ: ratio of performance to interquartile range; RER: range error ratio.

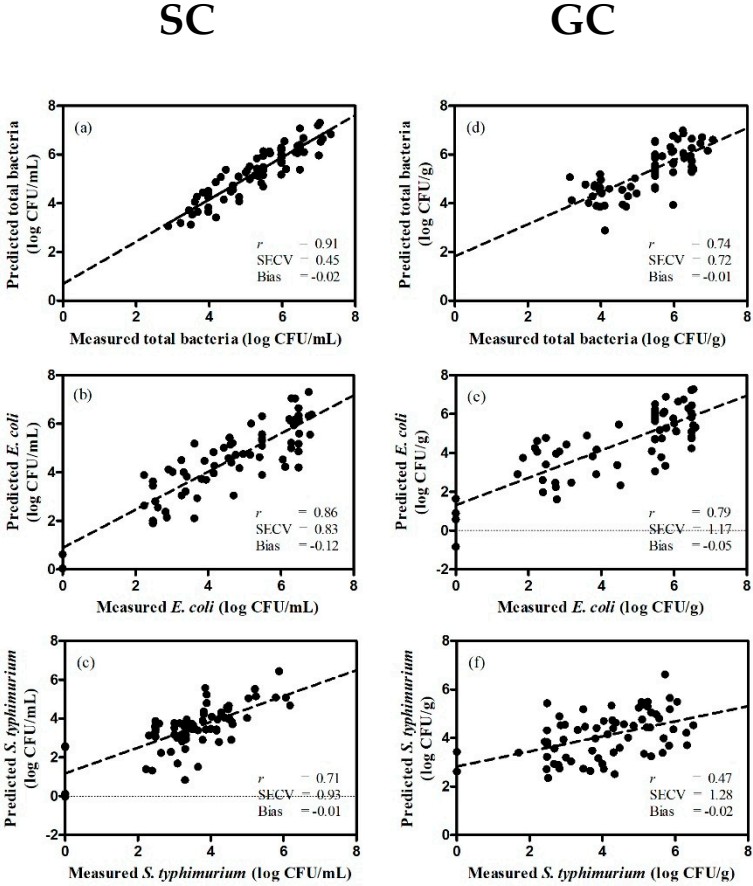

**Figure 4.** Comparison between actual measured and predicted values, using the full-cross validation of the PLS models for total bacteria, *E. coli*, and *S. typhimurium* contamination in the drained solution of shredded cabbage (SC, **left**) and mixed sample of ground cabbage and bacterial suspension (GC, **right**), measured using a portable NIR spectrometer (700–1100 nm).

The ratio of prediction of deviation (RPD) is a principle indicator for considering the possibility to use NIR. RPD below 1.5 indicates that the calibration was not useable; a value between 1.5 and 2.0 reveals a possibility to distinguish between high and low values, while a value between 2.0 and 2.5 makes it possible to make predictions in an approximate quantitative propose [11,28,29]. In our case, the SC analysis showed the highest RPD in total bacteria (2.44), followed by *E. coli* (1.95). On the other hand, for the GC, the RPD of *E. coli* was only at 1.61 and 1.44 of the total bacteria. For *S typhimurium*, the RPD of SC was 1.36, when it was 0.47 for the GC. According to these RPD values, the total bacteria from the SC samples could possibly be used as an approximate quantitative prediction. The detection of *E. coli* in both samples were considered to make it possible to distinguish between high and low counts.

From the r and RPD values, the detection of total bacteria from the SC was optimal, which is similar to the study by Suthiluk et al. [9], who reported using the SW-NIR through a PLS regression which was capable of prediction of total bacteria in the washing solution of shredded cabbage (*r* = 0.92, SEP = 0.46). Furthermore, *E. coli* detection in SC samples was possible for the separation between the high and low counts, which was correlated using NIR for classifying *E. coli* in a phosphate buffer saline [13]. The optimal prediction result was found in the *E. coli* concentrate, which was more than 4 log CFU·mL$^{-1}$. However, Kiefer et al. [30] reported that the spectra of *E. coli* concentrations below 5 log CFU·mL$^{-1}$ was not different. This implies that too low concentrations of bacteria are responsible for low feasibility of prediction. In this study, *E. coli* concentrations between 0 to 7 log CFU·mL$^{-1}$ were effective in distinguishing between high and low counts.

Moreover, the ratio of performance to inter-quartile range (RPIQ) was calculated for more supporting evidence. Bellon-Maurel et al. [31] recommended using IQ instead of SD for calculating RPD, which better represents the population spread. The statistically similar RPD should be based on the guidelines following Williams [32], who suggested that RPIQ values higher than 3 are useful for screening, that values greater than 5 can be used for quality control, and that values greater than 8 can be used for any application. In this study, the RPIQ values were higher than 3, but less than 5 with *E. coli* (3.45) and total bacteria (3.40) in the SC samples.

The range error ratio (RER) is a method for standardizing the RMSECV, by relating it to the range of the reference data. The RER value of total bacteria was 9.55 in the SC samples, while that of *E. coli* was 8.18. RER values between 7 and 20 are classified as a poor model and can only be used for screening purposes. However, RER values of *E. coli* in GC samples (5.67), total bacteria in GC samples (5.48), and *S. typhimurium* in both samples (SC = 6.72, GC = 5.10), were less than 6 and that was not recommended for any application. RER values of this study were based on the guidelines for interpretation of RER [33–35]. The results of RPIQ and RER were similar and used to predict the total bacteria and *E. coli* in SC, a fair model for the screening propose. This confirms that the use of SC samples to predict the total bacteria and *E. coli* content was suitable.

## 4. Conclusions

The results suggested that the SW-NIR could be applied to detect microbial contamination in shredded cabbage. Nevertheless, the pretreatment method is recommended for application on the original spectra, in order to improve the discrimination efficiency in the PCA and PLS models. The difference of the PLS model results between the drained solution (SC) and the mixed sample of ground cabbage and microbial suspension (GC). The efficiency in detecting microbial contamination was higher in the SC, as observed from the correlation coefficient (*r*) and obtained RPD. The transparency of the measured sample highly affected the quality of measured spectra. The cabbage cell composition, dispersed in the GC sample, could interfere with the spectra absorbance. Although the difference at consecutive incubation times could not be clearly seen in the results, the NIR technique demonstrates potential as a rapid method for detecting or monitoring microbial contamination in fresh-cut shredded cabbage. Nevertheless, further studies are required for more precise results in terms of the amount of contamination or the capability to differentiate the types of bacteria.

**Author Contributions:** Project administration, K.N.; Supervision, S.K., S.O. and K.N.; Writing–original draft, B.M.; Writing–review & editing, C.W.-A., P.P. and P.M.

**Funding:** Thailand Research Fund: PHD/0059/2552.

**Conflicts of Interest:** The authors declare no conflict of interest.

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
