# Peer review of "Feasibility of Determination of Foodborne Microbe Contamination of Fresh-Cut Shredded Cabbage Using SW-NIR"

_agriengineering, doi:10.3390/agriengineering1020018_

Round 1
Reviewer 1 Report
The authors used short wavelength near infrared to predict the quantity of total bacteria, Escherichia coli and Salmonella spp. on shredded cabbage and ground cabbage. They reported the prediction results of microbial spoilage were more reliable on shredded cabbage than ground cabbage.
The language is not acceptable and it should be improved by correcting all grammatical errors and typos. The authors should seek help from native English speakers or professional editing service.
There are very limited discussion based on the results.
Line 10: Hours should be replaced with h throughout the manuscript.
Line 69: Are there any strain numbers or designations for these two strains?
Line 107-112: The detection limit of direct plating should be described.
Author Response
Reviewer 1:
Line 10: Hours should be replaced with h throughout the manuscript
Reply: follow the suggestion by correcting the word throughout the manuscript
Line 69: Are there any strain numbers or designation of these two strains?
Reply: the strain numbers were added.
Line 107-112: The detection limit of direct plating should be described.
Reply: We added more detail of microbial analysis in the 2.5 session (Line 135-146)

Reviewer 2 Report
Interesting and useful study.
I have however some questions and suggestions:
-The aim of the study should be clearly stated at the end of the introduction.
-The number of samples is too small.
-PLS regression or PCA?
-Around 140 samples were used to calibrate the SW-NIR methodology, but how many samples were used to validate the model?
-In Table 1, are the results expressed per mL or per g? It is not clear... If the washing solution was analysed, it should be per mL...
-The conclusions are scarce, probably due to the low number of samples analysed.
-The model may be improved by adding more samples.
Author Response
Reviewer 2:
1. The aim of the study should be clearly stated at the end of the introduction.
Reply: The aim of the study was revised and located at the end of the introduction.
2. The number of samples is too small
Reply: Although it seemed too small of sample sizes, each evaluated spectrum was collected from an individual sample which mean many replicates of the experimental design.
3. PLS regression or PCA?
Reply: Models were developed using PLS regression by full cross validation.
4. Around 140 samples were used to calibrate the SW-NIR methodology, but how many samples were used to validate the model?
Reply: There was no external set for validation.
5. In table 1, are the results expressed per mL or per g? It is not clear. If the washing solution was analysed, it should be per mL.
Reply: The amounts of CFU per mL stand for the results from non-ground sample (SC) which was used of the washing solution to be analysed. On the other hand, the amounts of CFU per g stand for the results from ground sample (GC) which were mixed of water and the ground material.

Round 2
Reviewer 1 Report
It can be accepted for publication.
Author Response
Thank you very much for your effort reviewing this manuscript.
Reviewer 2 Report
The authors have corrected the manuscript following most of the reviewers' previous coments and suggestions.
Nevertheless, the authors should really use an independent set of samples to validate their calibration model. It is still possible to collect new data to do this, since this groups of samples should be independent.
Author Response
Reviewer 2:
The authors have corrected the manuscript following most of the reviewers' previous comments and suggestions.
Nevertheless, the authors should really use an independent set of samples to validate their calibration model. It is still possible to collect new data to do this, since this groups of samples should be independent.
Reply:
We totally agree with your suggestion to use an independent set of samples to validate the calibration model. We really wish to do that too for improving the quality of this article.
However, it quite impossible to conduct the additional experiment at the moment, firstly due to the availability of the equipment and secondly the time constraint.
The equipment that we used in this experiment, a portable NIR spectrometer (FQA-NIR GUN), is not available at the moment and it’s not our property. It belongs to another research center and needs a period of time to borrow it again. Besides, we also have to prepare two bacterial suspensions (E. coli and S. typhimurium) starting from ordering them from the culture collection in another research center. We don’t have them at hand now.
Therefore, we are afraid that it could not be finished within the period of revision time if we have to conduct the experiment. We do really appreciate your effort for reviewing this manuscript. We really hope that you will understand our condition. Could you please suggest us for other solutions for improving this manuscript?
